# Covariance of Marine Nucleocytoplasmic Large DNA Viruses with Eukaryotic Plankton Communities in the Sub-Arctic Kongsfjorden Ecosystem: A Metagenomic Analysis of Marine Microbial Ecosystems

**DOI:** 10.3390/microorganisms11010169

**Published:** 2023-01-09

**Authors:** Kang Eun Kim, Hyoung Min Joo, Taek-Kyun Lee, Hyun-Jung Kim, Yu Jin Kim, Bo Kyung Kim, Sun-Yong Ha, Seung Won Jung

**Affiliations:** 1Library of Marine Samples, Korea Institute of Ocean Science & Technology, Geoje 53201, Republic of Korea; 2Department of Ocean Science, University of Science & Technology, Daejeon 34113, Republic of Korea; 3Unit of Next Generation IBRV Building Program, Korea Polar Research Institute, Incheon 21990, Republic of Korea; 4Risk Assessment Research Center, Korea Institute of Ocean Science & Technology, Geoje 53201, Republic of Korea; 5Division of Polar Ocean Science Research, Korea Polar Research Institute, Incheon 21990, Republic of Korea

**Keywords:** *Aureococcus anophagefferens*, eukaryotic plankton communities, *Gyrodinium helveticum*, metagenomics, nucleocytoplasmic large DNA viruses, sub-arctic sea

## Abstract

Nucleocytoplasmic large DNA viruses (NCLDVs) infect various marine eukaryotes. However, little is known about NCLDV diversity and their relationships with eukaryotic hosts in marine environments, the elucidation of which will advance the current understanding of marine ecosystems. This study characterizes the interplay between NCLDVs and the eukaryotic plankton community (EPC) in the sub-Arctic area using metagenomics and metabarcoding to investigate NCLDVs and EPC, respectively, in the Kongsfjorden ecosystem of Svalbard (Norway) in April and June 2018. *Gyrodinium helveticum* (Dinophyceae) is the most prevalent eukaryotic taxon in the EPC in April, during which time *Mimiviridae* (31.8%), *Poxviridae* (25.1%), *Phycodnaviridae* (14.7%) and *Pandoraviridae* (13.1%) predominate. However, in June, the predominant taxon is *Aureococcus anophagefferens* (Pelagophyceae), and the NCLDVs, *Poxviridae* (32.9%), *Mimiviridae* (29.1%), and *Phycodnaviridae* (18.5%) appear in higher proportions with an increase in Pelagophyceae, Bacillariophyceae, and Chlorophyta groups. Thus, differences in NCLDVs may be caused by changes in EPC composition in response to environmental changes, such as increases in water temperature and light intensity. Taken together, these findings are particularly relevant considering the anticipated impact of NCLDV-induced EPC control mechanisms on polar regions and, therefore, improve the understanding of the Sub-Arctic Kongsfjorden ecosystem.

## 1. Introduction

Viruses are the most abundant organisms on earth, with the highest genetic diversity compared to other biological organisms in marine ecosystems [1,2]. In particular, viruses kill approximately 20% of the oceanic microbial biomass daily, thus, providing significant energy sources, including nutrients and dissolved organic matter [1,3]. Although little is known about viruses that infect eukaryotic plankton communities (EPCs), nucleocytoplasmic large DNA viruses (NCLDVs) are known to infect various eukaryotic organisms [4,5], and to be strongly associated with eukaryotic plankton [3,4,6]. In fact, the diversity and richness of NCLDV families, namely *Mimiviridae* (i.e., *Megaviridae*), *Phycodnaviridae*, *Pandoraviridae*, *Poxviridae*, *Iridoviridae*, *Marseilleviridae*, *Pithoviridae*, *Ascoviridae*, *Asfarviridae*, and *Mininucleoviridae*, are higher than those of other viruses in the sea [7,8]. Meanwhile, various marine autotrophic eukaryotes, such as haptophytes, chlorophytes, and dinoflagellates, act as hosts to NCLDVs, which can have fundamental roles in the killing or regulation of phytoplankton blooms in marine coastal waters [9]. Additionally, NCLDVs infect heterotrophic organisms [10,11]. Thus, the diverse groups of NCLDVs highlight the ecological impact exerted by the marine ecosystem in biogeochemical cycling [1,12,13,14]. In fact, several viruses belonging to *Phycodnaviridae* and *Mimivirdae* reportedly contribute to biological carbon export from marine surfaces to deep layers, via host-cell death [15,16]. Despite the ecological importance of NCLDVs, data related to the relationship between NCLDVs and eukaryotic hosts remains limited. 

Arctic warming, also known as ‘Arctic amplification’, is increasing rapidly compared to global warming in other continents; that is, temperatures have risen faster on the Arctic continent over the past decades [17,18]. Changes caused by ‘Arctic amplification’ include weak stratification, increased vertical mixing, and sea-ice loss. These changes in the Arctic Ocean are similar to those observed in the Southern Ocean, referred to as ‘Atlantification’ [19]. This continued influx of warm Atlantic waters into the Arctic Ocean will likely result in environmental changes caused by ice and snow melting, as well as the development of new ecosystems, including those inhospitable to Arctic endemic species [20,21]. The sub-Arctic Kongsfjorden (Svalbard, Norway) ecosystem is affected by Atlantic Ocean currents characterized by high temperatures and high salinity [22]. Payne and Roesler [22] explained that in Kongsfjorden, the intrusion of Atlantic Waters warmed subsurface waters by 4 °C in 15 years. This further led to sea ice melting, rapid glacial retreat, altered water-column structure, and changes in chlorophyll-*a* in the region. In addition, the Kongsfjorden Sea exhibits distinct differences in the Arctic Ocean ecosystem when the water temperature drops below 0 °C in April, marking the beginning of the white nights. Meanwhile, the highest primary productivity is observed in June. In this season, the pre-dominant phytoplanktons include dinoflagellates, diatoms, Chrysophyceae, and Dictyochophyceae [23,24]. Furthermore, Coupel, et al. [25] and Fujiwara, et al. [26] observed an increased appearance of nano- and pico-phytoplankton, such as cryptophytes, including *Micromonas* and cryptophytes species. Until recently, it was believed that the increase in phytoplankton primary production within the ice-covered Arctic Ocean only began after the sea ice receded, when the water mass stratified to provide sufficient light for photosynthesis [27,28]. However, in certain areas of the Arctic, earlier phytoplankton blooms may have been induced by early sea-ice retreat, occurring widely throughout the Arctic [29]. Consequently, the Arctic Ocean experiences weakened ocean stratification with the bottom layers susceptible to surface heat by enhanced vertical mixing and lower salinity caused by ice melting [30]. However, a comprehensive understanding of Arctic phytoplankton indicator dynamics remains elusive. Moreover, the relationship between NCLDVs and phytoplankton dynamics in the Kongsfjorden ecosystem is unknown.

Recently, a metagenomic tool for analysis of the virus community has been used in various marine ecosystems, including pole-to-pole and deep-sea ecosystems [6,31]. De novo sequencing and bioinformatics technologies have been developed using NCLDV data obtained through sequences extracted from environmental samples collected during various ocean research cruises [32,33,34]. The rapid advancement of metagenomic technology has facilitated the facile acquisition of high-resolution sequences, further enabling the analysis, and more accurate classification, of microorganisms at the genus and species levels, within populations and/or communities in ecological studies. Moreover, owing to advancements in metagenomics, a technology has been developed to analyze microbial diversity using a smaller volume of seawater and DNA [35]. Hence, the current study seeks to examine the NCLDVs and EPC in a sub-Arctic Kongsfjorden ecosystem (glacial fjord) by employing these metagenomic platforms. Although Kongsfjorden, an Arctic glacier fjord area on the western coast of the Spitsbergen Island in Svalbard, is geographically situated in the Arctic, it is affected by various factors, including the Atlantic Ocean currents and Arctic Sea [22,36,37]. In particular, the active physical properties are influenced by changes between the Svalbard coastal waters (glacial outflow) and the western continental shelf coastal waters [38]. Moreover, the surface waters of Kongsfjorden vary with complex factors, including tides, currents, wind, precipitation, water inflow from glacial melt and the coast, and sea ice/glacier drift [39]. The primary aims of this study are to (1) elucidate the spatial distribution and comparison between NCLDVs and EPCs during the early white night (April) and mid-summer (June); (2) identify NCLDVs with a strong association and co-occurrence with the EPCs; and (3) describe potential NCLDV hosts. Collectively, these findings will aid future studies of viral and host community structures in the Arctic seas, particularly within cold waters near the Arctic Ocean, to provide an explicit linkage between viruses, host cells, and water masses and currents.

## 2. Materials and Methods

### 2.1. Measurement of Environmental Parameters

The temperature and salinity in the sub-Arctic Kongsfjorden ecosystem were measured using the Sea-Bird 911 plus CTD model (Sea Bird Electronics, Inc., Bellevue, WA, USA). Water samples were collected using a 20-L rosette-type Niskin sampler bottles (Ocean Test Equipment, Inc., Ft. Lauderdale, FL, USA) from the surface (1 m below the sea surface) and bottom layers (1 m above the bottom sediment) at seven sampling sites (St. 1, 2, 4, 5, 7, 8, 9 and 10) on board the RV TEISTEN in April 2018 and six sites (St. 1, 2, 5, 7, 8 and 10) on board the RV MS-FARM in June 2018 (Figure 1 and Appendix A). The bottom layers (St. 5, 7, 8, and 10) were investigated in June at a depth of 100 m due to mechanical issues with the Niskin sampler of the research vessel. The size-fractioned chlorophyll-*a* was analyzed according to three continuous steps: >20 μm (micro-), 2–20 μm (nano-), and 0.2–2 μm (pico-sized chlorophyll-*a*). Next, 1000 mL of seawater was filtered sequentially through 20, 2, and 0.2 μm pore size and 47 mm diameter Merck Millipore filters (NY2004700, TTTP04700, and GTTP04700, respectively). Gravity filtration was applied for the 20 μm filtrations, while a vacuum of approximately 200 mm Hg was used for the 2 and 0.2 μm filtrations. Each size-fractionated filter was soaked in 4 mL of cold 90% acetone-distilled water (*v*/*v*). Subsequently, chlorophyll-*a* was extracted for 24 h at 4 °C in the dark, and its concentration was measured using a fluorometer (Trilogy; Turner Designs, Sunnyvale, CA, USA).

### 2.2. Metavirome Analysis of NCLDVs

To analyze the dsDNA virus community, 30 L of seawater was collected for duplicated experiments of 15 L each from the surface and bottom layers in each sampling site and stored at approximately 4 °C for transportation to the DASAN Arctic Research Center of Korea Polar Research Institute, the Republic of Korea (located in Kongsfjorden, Svalbard, Norway). However, some water samples (St. 1, 5, 7, 8, and 10) collected in April were not sufficient for duplicate experiments, owing to a disturbance caused by the sea-ice movement and large waves during sampling.

dsDNA virus communities were analyzed according to a previously described protocol [34] with certain modifications. To extract genomic DNA (gDNA) from the dsDNA virus community, 15 L of seawater were passed through a 3-μm filter (TSTP04700, 47 mm, Merck Millipore, Bedford, MA, USA) to remove inorganic and organic particles. DNA viruses were harvested from pre-filtered seawater by flocculation, filtration, and resuspension methods using FeCl_3_, as previously described [40]. To aggregate DNA viruses onto Fe_3_^+^ ions, the reservoir (cylindrical with black-colored acrylic materials) was maintained at 20 °C for 1 h. The seawater in the reservoir was mixed with three impellers, each 6 cm wide and 15 cm long, at 10 rpm [41]. The aggregated DNA viruses flocculated using Fe_3_^+^ ions were harvested using a 0.2 μm polycarbonate membrane (111106; 47 mm; Whatman, Buckinghamshire, UK) and stored at −80 °C until total gDNA extraction. Although the temperature for resuspending the virus community aggregated in Fe ions on the filter should be maintained at 4 °C [40], despite the losses of viral DNA, they were kept at −80 °C due to sample transportation from Norway to South Korea. Subsequently, the membrane was cut into eight pieces and placed in a 50-mL conical tube with a suspension buffer (10 mL of 0.1 M EDTA, 0.2 M MgCl_2_, 0.2 M Ascorbate). Flocculated viruses attached to the membrane were suspended and mixed in the buffer (pH 6, adjusted using approximately 5 mL of 10 M NaOH). Total gDNA was extracted using a Viral Gene-spin Viral DNA/RNA Extraction Kit (iNtRON Biotechnology, Seoul, Republic of Korea). The gDNA library for metagenomics was prepared using the NEBNext Ultra II DNA Library Prep Kit (Illumina, San Diego, CA, USA). The sequencing library was prepared through random fragmentation of the DNA sample, followed by 5′ and 3′ adapter ligation. The adapter-ligated fragments were amplified using polymerase chain reaction (PCR) over 12 cycles (denaturation: 98 °C 30 s [1 cycle]; annealing: 98 °C 10 s, extension: 65 °C 75 s [12 cycles], final extension: 65 °C 5 min [1 cycle]). Illumina tagged Universal primer (5′-AAT GAT ACG GCG ACC ACC GAG ATC TAC ACT CTT TCC CTA CAC GAC GCT CTT CCG ATC-3′) and Indexed primers using the NEBNext Multiplex Oligos (Illumina, San Diego, CA, USA). The DNA library was then sequenced via paired-end sequencing using the Illumina HiSeq 2500 platform.

Subsequently, the modified bioinformatics analysis methods were performed in accordance with the protocol described by Kim et al. Kim, Jung, Park, Kim, Lee, Ha, and Lee [34]. Briefly, the raw sequencing data (Fastq file) was sorted by trimming using the Qiagen CLC genomics workbench (v. 20.0.4 Qiagen, Hilden, Germany), which removed low-quality adapters according to the criteria of two ambiguous nucleotides and a minimum length of 50 nucleotides. Next, an assembly analysis was performed using metaSPAdes v. 3.13.0 [42] and five k-mer sizes (K-mer: 21, 33, 55, 77, 99, 128) were selected. A quality check of viral contigs was then performed using Check V v.1.0 to remove contiguous host regions and non-viral regions from the assembled proviruses [43]. Through the Check V quality check, only virus contigs of medium completeness or higher, Read mapping was applied to BBMap v38.51 [44] using 95% minimum alignment identity (Appendix A). The virus contigs quality-checked through Check V and Read mapping were subjected to virus taxonomy analysis using Basic local alignment search tool (BLASTn) analysis. BLASTn was performed using the Microbial Genomic Module in the CLC Genomics Workbench settings with the following option: NCBI Viral RefSeq database. The results were sorted according to virus taxon (eukaryote viruses and bacteriophages) and NCLDVs in eukaryote viral communities using the customized taxonomic profiling assignment coding developed by Kim et al. Kim, Jung, Park, Kim, Lee, Ha, and Lee [34] in the open-source software R (version 3.6.3, R Core Team, 2020) within R studio (version 1.3.959).

### 2.3. Metabarcoding Analysis of Eukaryotic Plankton Communities (EPCs)

Our laboratory’s methodology was employed to perform the metabarcoding analysis of EPCs [45,46]. A 0.2 μm polycarbonate filter (GTTP04700; 47 mm) was used to analyze EPC from 2 L of seawater collected from sample sites consistent with NCLDV samples. Distilled water (50 mL, approximately 50–60 °C) was used to wash the harvested filter three times to remove environmental DNA debris [45]. The filters of each fraction were cut into several pieces using sterilized scissors before gDNA extraction. The gDNA was extracted using the DNeasy PowerSoil Kit (Qiagen, Valencia, CA, USA). Amplicon sequencing was performed using Illumina tagged forward (TAReuk454FWD1) and reverse (TAReukREV3) primers of the 18S rRNA gene (V4–V5 regions) (Appendix A). The products from the first PCR were amplified using tags of the Nextera XT 96 index kit v2 (Illumina). The final PCR products were purified using a QIAquick PCR purification kit (Qiagen, Hilden, Germany), and DNA concentration was measured using a Bioanalyzer 2100 (Agilent Technologies, Palo Alto, CA, USA). All samples with equal concentrations of the amplified products were mixed, and the pooled samples were analyzed using a Mi-Seq platform (Illumina).

Bioinformatics analysis from raw data (FASTQ files) using Mi-Seq equipment was performed according to the Bioinformatic process [46] and pre-processed using the Mi-Seq Control Software (MCS) v2.4.1 and FastQC [47]. Further, the quality score distribution per base and poor-quality sequences were flagged. Ambiguous and chimeric reads were removed, and noised sequences (denoising), containing operational taxonomic units (OTUs) with 1, 2, or 3 reads, were removed at a cut-off of 97%. Fast length adjustment of short reads (FLASH) was used to merge paired-end reads [48]. To remove short sequence reads (<150 bp), low-quality sequences with a score < 33, singletons, and non-target sequences, we performed a quality check using BLAST [49]. Subsequently, the sequences were compared with reference sequences from the NCBI database. Based on E-values < 0.001, a similarity score of >97% was used to classify the sequences. The CD-HIT-OTU software [50] was used for clustering and analyzing the OTUs.

### 2.4. Statistical Interpretation of the Data

To explain relationships between common taxa of NCLDVs and EPCs (each displaying a relative abundance > mean 1%), Spearman’s correlation analysis was performed using SPSS v.12 (SAS Institute Inc., Cary, NC, USA), and the significant correlation coefficient was expressed as a circular flow chart. A cluster analysis was performed in the heatmap to express differences between common OTUs of NCLDVs and EPCs using the Bray–Curtis dissimilarity method and the group average. The circular flow chart and heatmap were plotted using the ‘ggplot2′ package [51] with a combination of custom R and R Studio (v. 1.2.5042). Using the ranked similarity matrix, an ordination plot was produced through non-metric multidimensional scaling (NMDS) using PRIMER 6. Hierarchical agglomerative clustering using the group average method was performed on the most abundant OTUs based on the groups selected from the NMDS analysis. A redundancy analysis (RDA) was performed using the CANOCO program version 4.55 to estimate the common NCLDVs, EPCs, and environmental relationships [52]. All axes were tested for significance using 999 unrestricted Monte Carlo permutations. The extended local similarity analysis (eLSA) was performed using analyzed parameters (common taxa of NCLDVs and EPC, and environmental factors, such as water temperature, salinity, and size-fractionated chlorophyll-*a* concentration) [53]. The *p*-value was calculated using permutation testing while ensuring accuracy, and the Q-value was determined to estimate the likelihood of false positives [54]. Network visualization was performed using Cytoscape v3.9.2 [55]. Here, the *p*- and Q-values were <0.05, and the Pearson correlation coefficients (PCC) were set to >0.5. The time lag was set to zero as the analyzed data did not include time series data [56].

## 3. Results

### 3.1. Changes in Environmental Factors in the Sub-Arctic Kongsfjorden

We found the mean salinity and water temperature in April to be 30 and 0 °C, respectively, in their vertical distributions (Figure 2a,b). The chlorophyll-*a* concentration in April was uniformly distributed at <0.1 μg L^−1^ across all measured data (Figure 2c). In June, the mean water temperature increased to 2 °C, while that of salinity decreased to 28 in the surface layer, implying that the salinity of the surface layer was affected by sea ice or glacial meltwater. The chlorophyll-*a* concentration in the surface layer adjacent to the inner bay (St. 1) was relatively high at 1.2 μg L^−1^, however, at the other sampling sites was <0.3 μg L^−1^.

### 3.2. Changes in NCLDVs in the Sub-Arctic Kongsfjorden

Taxonomic analysis of contigs in the genome fragments confirmed their identity as dsDNA viruses. A total of 15,727 DNA virus contigs and 2138 contigs affiliated with NCLDVs were detected via metavirome analysis. More specifically, in April and June, 837 and 967 mean NCLDVs contigs were identified, respectively. The NCLDVs were classified into three groups at 64% similarity using NMDS analysis (Figure 3). Virus(v)Group 1, associated with the ‘NCLDVs in April’, comprised *Mimiviridae* (31.8%), *Poxviridae* (25.1%), *Phycodnaviridae* (14.7%), *Pandoraviridae* (13.1%), and *Iridoviridae* (7.4%). The predominant virus was virus Operational Taxonomic Unit (vOTU) #075 (*Mimivirus terra2*, *Mimiviridae*) at a relative abundance of 7.11%, followed by *Acanthamoeba polyphaga moumouvirus* (vOTU#002, *Mimiviridae*) at 5.99% (Figure 4a and Appendix A). vGroup 2 was associated with the ‘NCLDVs in April collected from certain surface layers (St. 1, and 5)’; *Phycodnaviridae* accounted for 20.5% of the NCLDV population. vGroup 3 was associated with the ‘NCLDVs in June’, wherein *Poxviridae*, a pre-dominant group, contributed 31.0%, which was higher than that in April. Meanwhile, *Pandoraviridae* rapidly decreased to 62% compared to April. Meanwhile, in June, most viruses were evenly distributed between the surface and bottom layers.

The NCLDVs in April and June comprised 413 taxa and 437 taxa, respectively, of which, 78.1% (411 taxa) were common to both sampling time points. However, only 28 NCLDV taxa with a mean relative abundance > 1% were common in all samples and overlapped by 10% between April and June (Figure 4b). Of these common NCLDV taxa, 17 were relatively increased in April compared to June, while 11 taxa decreased in June. In particular, *Mimivirus terra2* (vOTU#75, *Mimiviridae*), *Pandoravirus neocaledonia* (vOTU #102, *Pandoraviridae*), *Pandoravirus macleodensis* (vOTU #101, *Pandoraviridae*), and *Cotia* virus SPAn232 (vOTU #29, *Poxviridae*) rapidly decreased in June, while *Acanthamoeba polyphaga moumouvirus* (vOTU#17, *Mimiviridae*), *Orpheovirus* IHUMI-LCC2 (vOTU#90, *Phycodnaviridae*), *Megavirus chiliensis* (vOTU#2, *Mimiviridae*), *Chrysochromulina ericina* virus (vOTU#26, *Mimiviridae*), and *Emiliania huxleyi* virus (vOTU#39, *Phycodnaviridae*) rapidly increased in June.

### 3.3. Dominant Eukaryotic Plankton Operational Taxonomic Units

The mNGS results for the EPCs are summarized in Appendix A. Sequences and read counts (97% cut-off) were generated across the EPCs at means of 73,258,358 and 174,478 (61,793), respectively. A total of 218,599 (87,698) reads were detected in June, which was approximately twice that in April. Moreover, the eukaryotic protist OTUs (pOTUs) of the EPCs in April and June comprised 448 taxa and 144 taxa, respectively. Of these, 18.9% (94 taxa) of the pOTUs were common to both sampling time points. The unique pOTUs in April and June represented 71.1% (354 taxa) and 10.0% (50 taxa), respectively. In particular, the common pOTUs (with a mean relative abundance > 1% in all samples) overlapped in 30 taxa (60.0%) over the two months (Figure 4b). The EPCs were classified into two groups at 64% similarity using NMDS analysis (Figure 3b). protist(p)Group 1 was associated with the ‘EPC in April’, comprising Dinophyceae (40.7%), Arthropoda (8.7%), Pelagophyceae (8.1%), Chlorophyta (5.9%), and Bacillariophyceae (5.2%), and appeared to be evenly distributed at all sites. Moreover, 11 pOTUs were common species; *Gyrodinium helveticum* (pOTU#7; 37.2%), *Aureococcus anophagefferens* (pOTU#0; 8.1%), uncultured copepod (pOTU#3; 5.7%), and *Pyramimonas australis* (pOTU#5; 3.7%) predominating, with an accumulated proportion of 53.7% (Figure 4a and Appendix A). The relative abundance of *Aureococcus anophagefferens* (pOTU #0) was high in the surface layer of St. 1 (30.7%). *Gyrodinium helveticum* was the dominant species in April, but not in June. Thus, the relative abundance of Dinophyceae appeared to be the highest in April. pGroup 2 was associated with the ‘EPC in June’, comprising Pelagophyceae (43.2%), Bacillariophyceae (15.3%), Dinophyceae (15.1%), Chlorophyta (11.4%), and Arthropoda (9.8%). Compared to those in April, EPCs in June exhibited a highly variable distribution. Moreover, the abundance of Chlorophyta and Pelagophyceae were 7.6% and 13.5% in the surface layer and 4.30% and 2.64% in the bottom layer, respectively, indicating a relatively high abundance in the surface layer. In this group, *Aureococcus anophagefferens* was a pre-dominant taxon, contributing 43.24%, which was a rapid increase from that observed in April and was evenly distributed regardless of the layer. *Thalassiosira* sp. (pOTU#22; 14.6%, Bacillariophyceae), *Heterocapsa rotundata* (pOTU#26; 11.7%, Dinophyceae), *Pyramimonas australis* (pOTU#5; 11.1%, Chlorophyta), and *Calanus finmarchicus* (pOTU#1; 6.9%, Arthropoda) were dominant with an accumulated proportion of 87.5%.

### 3.4. Potential Association between NCLDVs and the EPC

Bivariate Pearson correlations were performed to assess the correlations between common NCLDVs and EPC lineages with a mean relative abundances of >1%. Using this threshold (*p* < 0.05), 48 common NCLDV–EPCs lineage pairs were identified with strong significant positive correlation coefficients (*r* > 0.5, Figure 5, Appendix A). The NCLDV lineages were significantly correlated with a known host group of Dinophyceae (*r* = 0.65, *p* = 0.01). Among the NCLDVs, five *Mimiviridae* vOTUs (*Mimivirus terra2* [vOTU#75], *Megavirus chiliensis*, [vOTU#69], *Aureococcus anophagefferens* virus, [vOTU#12], *Cafeteria roenbergensis* virus BV-PW1 [vOTU#19], and *Chrysochromulina ericina* virus [vOTU#26]) was significantly correlated with nine EPC taxa (*Aureococcus anophagefferens* [pOTU#0], *Calanus finmarchicus* [pOTU#1], *Nephtys pente* [pOTU#11], *Chaetoceros* sp. [pOTU#13], *Polarella glacialis* [pOTU#15], *Gyrodinium helveticum* [pOTU#7], Uncultured Prasinophyceae [pOTU#20], Copepoda environmental [pOTU#3], Uncultured eukaryotes [pOTU#6]). One vOTU of *Phycodnaviridae* (*Emiliania huxleyi virus* 86 [vOTU#39]) and five vOTUs of *Pandoraviridae* (*Pandoravirus inopinatum* [vOTU#100], *Pandoravirus macleodensis* [vOTU#101], *Pandoravirus salinus* [vOTU#104], *Pandoravirus neocaledonia* [vOTU#102], and *Pandoravirus dulcis* [vOTU#99]) were significantly correlated with two (*Thalassiosira* sp. [pOTU#13] and *Pyramimonas australis* [pOTU#5]) and seven EPC pOTUs (*Chaetoceros* sp. [pOTU#13], *Navicula cari* [pOTU#9], *Gyrodinium helveticum* [pOTU#7], *Polarella glacialis* [pOTU#15], uncultured Prasinophyceae [pOTU#20], uncultured copepod [pOTU#3], and uncultured eukaryote [pOTU#6]), respectively. Two vOTUs of *Poxviridae* (*Melanoplus sanguinipes entomopoxvirus* [vOTU#70] and *Choristoneura biennis entomopoxvirus* [vOTU#240)] were significantly correlated with three pOTUs (*Heterocapsa rotundata* [pOTU#26], *Karlodinium venefium* [pOTU#23], and *Calanus finmarchicus* [pOTU#1]). Other families, including one vOTU of *Pithoviridae* (*Pithovirus sibericum* [vOTU#117]) and one vOTU of *Asfaviridae* (African swine fever virus [vOTU#6]) were each significantly correlated with one pOTU (uncultured Prasinophyceae [pOTU#20] and *Oithona similis* [pOTU#4]), respectively.

The RDA biplot in Figure 6 and Appendix A was analyzed to explain the relative importance of the variables between the NCLDVs and EPCs (F value: 5.73, *p* = 0.0001). The NCLDV families-common EPC, including environmental relationships represented in the plot, included six common families and 14 common NCLDV taxa and 21 common variables, including EPC phylum (or class level) and environmental factors that accounted for 43.4% (first axis) and 18.9% (second axis). More specifically, *Asfarviridae, Mimiviridae, Pandoraviridae*, and *Pithoviridae* were primarily distributed with salinity in April. In particular, most Pandoraviruses were strongly associated with *Gyrodinium helveticum* and Dinophyceae. In June, *Phycodnaviridae* and *Poxviridae* formed a cluster of sampling sites (Figure 6). Moreover, specific NCLDVs (particularly *Poxviridae* and *Phycodnaviridae*) were associated with each size-fractionated chlorophyll-*a* concentration, temperature, and Pelagophyceae. The two clusters reflect the succession of other phytoplankton communities that respond to the distinguished NCLDVs.

Network analyses between NCLDVs and EPCs revealed specifically associated co-occurrences. The 32 nodes and 57 edges exhibited a significant correlation between NCLDVs, the EPC, and environmental factors (Figure 7, Appendix A). A total of six vOTUs (one *Mimiviridae*, four *Pandoraviridae,* and one *Poxviridae*) were highly correlated with four specific pOTUs (two Chlorophyta, one Annelida, and one Dinophyceae) and four environmental factors (salinity, dissolved oxygen, and pico- and nano-sized chlorophyll-*a* concentrations). More specifically, *Pandoravirus inopinatum* (vOTU#100, *Pandoraviridae*) and *Pandoravirus salines* (vOTU#104, *Pandoraviridae*) exhibited co-occurrence with Uncultured eukaryotes (pOTU#6) and salinity, respectively (Figure 7). Meanwhile, *Mimivirus terra2* (vOTU#075, *Mimiviridae*) was significantly and positively correlated with *Polarella glacialis* (pOTU#15, Dinophyceae) and Uncultured Prasinophyceae (pOTU#20, Chlorophyta).

## 4. Discussion

NCLDVs infect numerous autotrophs, including haptophytes, chlorophytes, and pelagophytes [5,57]. The current study aimed to elucidate how the NCLDV community changes with the increasing diversity and abundance of the EPC. We found that a variety of NCLDVs had co-occurrence with a ubiquitous host or that an NCLDV was significantly correlated with various hosts. The findings of the current study provide novel insights to help elucidate the significant relationships and ecological processes that exist between NCLDVs and EPCs in the context of ecological emergence, and how NCLDVs contribute to ecological functions in the sub-Arctic Kongsfjorden.

The NMDS analysis showed apparent differences in each NCLDV and EPC composition between early white night and mid-summer. These changes occurred according to environmental changes, namely warm water temperature influx and increased light intensity. Endo et al. [9] reported that NCLDV families correlate with specific eukaryotes and infect many eukaryotic lineages. Herein, *Mimiviridae* viruses were the most commonly detected compared to other NCLDV families, which agrees with the results of a previous study [57]. This is likely due to the diversity of *Mimiviridae* structure and, a subsequent, broad range of hosts and environments compared to the reference sequences of known viral genomes [58]. In particular, *Mimiviridae* species are frequently found in low-temperature seawater, such as in the Arctic [59,60], which is likely related to the frequency and types of hosts. Indeed, our results showed that *Mimiviridae and Pandoraviridae* were significantly and positively correlated with specific EPC taxa, namely, *Aureococcus anophagefferens* (pOTU#0) increased rapidly in June, as did *Megavirus chiliensis* (vOTU#69) and *Chrysochromulina ericina virus* (vOTU#26) also increased. *Chrysochromulina ericina virus* was predicted to be an abundant viral group as it readily infects microalga (the haptophyceae *Haptolina ericina*, formerly *Chrysochromulina ericina)* that are widespread marine plankton species [61,62]. Although no significant correlation was observed among *Aureococcus anophagefferens* (pOTU#0), *Pyramimonas australis* (pOTU#5) and *Aureococcus anophagefferens virus* (vOTU#12), *Orpheovirus* IHUMI-LCC2 (vOTU#90), and *Emiliania huxleyi* virus 86 (vOTU#39), as the hosts increased, these viruses also increased, becoming the common taxa in June. Meanwhile, when *Gyrodinium helveticum* increased rapidly in April, specific Pandoraviuses (*Pandoravirus neocaledonia* [vOTU#102], *Pandoravirus macleodensis* [vOTU#101], *Pandoravirus dulcis* [vOTU#99]) also significantly increased. These viruses showed strong correlations with *Gyrodinium helveticum* (pOTU#7, Dinophyceae), which is commonly found in cold seawater, such as the Barents Sea, Canadian Arctic, and Greenland Sea [62,63,64]. Moreover, *Poxviridae* of *Choristoneura rosaceana entomopoxvirus* ‘L’ (vOTU#151) and *Melanoplus sanguinipes entomopoxvirus* (vOTU#70) increased with the increase in *Thalassiosira* sp. (pOTU#22), *Heterocapsa rotundata* (pOTU#26), and some copepods indicating that NCLDVs are involved in regulating the plankton community. Members of the family *Poxviridae* can infect a diverse range of vertebrates and invertebrates, although some poxviruses have narrow host ranges and others have very broad host ranges [65]. Copepods are among the most common invertebrates in Arctic sea ice [66]. Although, infected activities between *Poxviridae* and copepods have been few studies, our results in this study (e.g., the correlation between *Calanus finamchicus* and members of *Poxviridae*) will serve as a basis for future infection experiments.

The nano- and pico-sized fractioned chlorophyll-*a* concentration were also significantly correlated with the NCLDVs, which was consistent with the change in pico-sized *Aureococcus* and *Micromonas*. Milligan and Cosper [67] isolated the species-specific infection phage-like virus during a *Aureococcus anophagefferens* bloom, and Garry, et al. [68] identify its lytic characteristics of *Aureococcus anophagefferens* virus-1 (AaV-1) and strong infectivity against *A. anophagefferens*. Vardi, et al. [69] demonstrated that a viral infection controls *Emiliania huxleyi* blooms in artificial ecosystems. When light intensity and blooming density increase, virus activity also increases, potentially leading to bloom collapse [70]. Hence, in an extended interpretation, *Mimiviridae* may be an essential regulator of biochemical processes derived from phytoplankton.

Although it is well known that algal species are susceptible to infection by a specific NCLDV [32], recent studies suggest that NCLDVs are regulators of algal populations [60,71]. Here, most NCLDVs were found to be ubiquitously distributed throughout the study area. For example, *Mimiviridae*, *Poxviridae*, *Phycodnaviridae,* and *Pandoraviridae* accounted for a significant proportion of NCLDVs in the Kongsfjorden ecosystem. In particular, *Mimiviridae* and *Pandoraviridae* were abundant and were significantly correlated with pOTUs of Chlorophyta and Dinophyceae. However, other studies have reported that representative viruses exhibit host infection specificity. For example, *Heterocapsa circularisquama* DNA virus (HcDNAV) and *Emiliania huxleyi* virus infect the blooms of *H. circularisquama* and *E. huxleyi*, respectively [69,72]. In contrast, specific viruses have lineages of widespread host infection [73]. Our results indicate that common NCLDVs may regulate common host groups. The RDA results showed that NCLDVs were broadly associated with EPC. Endo et al. [9] explained that the distribution patterns of NCLDVs highlight their close interplay with various EPCs. The network analysis potentially represents relationships such as viral lysis, grazing, or other interactions. Indeed, some studies have demonstrated a significant association through network analysis between the host–NCLDV [46,74]. In particular, the network results in our study revealed that *Pyramimonas australis* (pOTU#5, Chlorophyta) significantly co-occurred with *Pandoravirus dulcis* (vOTU#99, *Pandoraviridae*), Uncultured Prasinophyceae (pOTU#20, Chlorophyta) co-occurred with *Mimivirus terra2* (vOTU#75, *Mimiviridae*). Meanwhile, previous network analysis revealed that *Pandoravirus* and the harmful algal blooms, *Akashiwo sanguinea* (Dinophyceae), exhibit significant co-occurrence [46]. Moreover, *P. macleodensis* and *P. salinus* primarily infect amoebas [75]. However, additional studies are needed to evaluate these hosts as substantial evidence has not been provided to conclude whether Pandoraviruses can infect *Gyrodinium* sp. or whether the host infectivities of Pandoraviruses are diverse because of given the high genetic similarity between Pandoraviruses and Phycodnaviruses [76]. Additionally, in co-occurrence and host prediction analysis performed by Kaneko, et al. [58], specific *Mimiviridae* viruses were found to be associated with Prymnesiales and Phaeocystales hosts (Haptophyta), while *prasinoviruses* and putative *prymnesioviruses* were strongly associated with the Prymnesiales host lineages.

Although the study of NCLDVs in marine ecosystems differs considerably from laboratory-based viral infection and isolation assays, co-occurrence analyses, such as network analysis, will provide important insights regarding inter-relationships in ecosystems. Therefore, we believe that the emergence of any viral community will influence host distribution depending on the infectious activity. Furthermore, we suggest that the viral community structure and infectious activity reflect host distribution [77]. The observation that NCLDV communities were mostly similar in the bottom and surface layers, suggests the presence of vertical NCLDV connectivity, implying that the absence of stratification may connect the surface and bottom NCLDV communities due to low water temperatures. Moreover, surface viruses and energy sources may be transferred by gravitational export to the bottom layers as sinking aggregated organic particles, including phytoplankton cells [78,79]. When organic matter in the surface layer is transported to the bottom layer, it is implied that viruses also interact with their hosts in the lower ecosystem [80]. Indeed, a recent study revealed that various *Mimiviridae* and *Phycodnaviridae* viruses could potentially infect their host species in the bottom layers through supplementation with biological carbon pumps [58,81]. A “viral shunt” refers to the lysis, and subsequent transformation, of the host cell into organic matter. According to the viral shunt pathway [82], viruses infect auto- and heterotrophs producing released substrates, such as dissolved organic matter and particulate organic matter, by host cell destruction. Viruses can then be transplanted into the deep layers of the ocean either by adsorption by host cells or infection of host cells [83,84]. Therefore, the vertical connectivity of various NCLDVs may be associated with the infected hosts in the bottom layer via vertical sinking.

## 5. Conclusions

This study describes the diversity of NCLDVs and EPC during the early white night (April) and mid-summer (June) in the Sub-Arctic Kongsfjorden ecosystem. In addition, the potential host lineages were described including the co-occurrence of various NCLDVs with a ubiquitous host, or the correlation between a single NCLDV and multiple hosts. NCLDVs and EPC populations were found to be similar between surface and bottom layers, however, differed between April and June sampling points. In particular, three *Phycodnaviridae*, two *Poxviridae*, three *Pandoraviridae,* and two *Mimiviridae* viruses accounted for an essential proportion of NCLDVs diversity. Furthermore, *Pandoraviridae* and *Mimiviridae* were strongly associated with Dinophyceae and Chlorophyta hosts, respectively. Additionally, *A. anophagefferens* (pOTU#0) increased as *Chrysochromulina ericina* virus (v#OTU26) increased, while *Gyrodinium helveticum* (pOTU#7) increased as *Pandoravirus neocaledonia*(v#OTU102)*, Pandoravirus macleodensis* (v#OTU102), and *Pandoravirus dulcis*(v#OTU101) increased. Although a correlation was not detected between *A. aophagefferens* and *Mimiviridae*, this family of viruses may contribute to the regulation of *A. ophagefferens* blooms as *Mimiviridae* is abundant in the Sub-Arctic Kongsfjorden ecosystem. Given the wide range of viral host species, not all marine viral hosts are defined. Nevertheless, our findings underscore the importance of NCLDVs as a component of marine microbial communities and refine our knowledge related to marine ecosystems. In conclusion, our findings, which are particularly relevant considering the anticipated impact of biological EPC control mechanisms by NCLDVs on polar regions, can contribute to a better understanding of the Sub-Arctic Kongsfjorden ecosystem. Future studies of both viral and host community structures in the Arctic Seas, particularly from the cold waters near the Arctic Ocean, could provide a more explicit linkage between viruses, host cells, and water masses and currents.

## Figures and Tables

**Figure 1 microorganisms-11-00169-f001:**
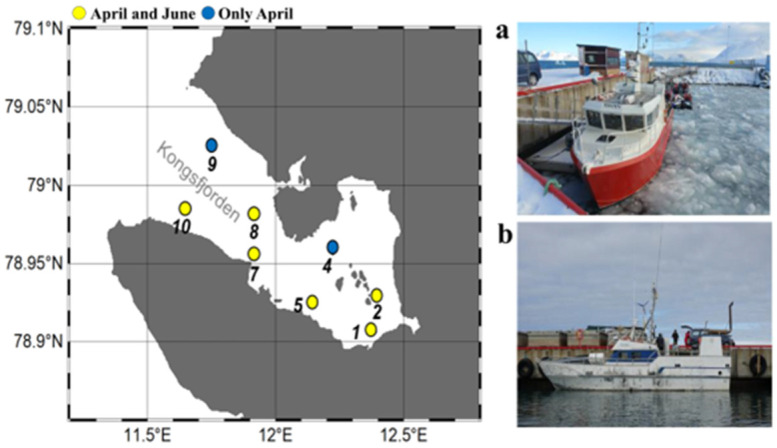
Sampling locations in the Sub–Arctic Kongsfjorden in April and June 2018. A total of eight sampling sites (St.) were investigated at St. 1, 2, 4, 5, 7, 8, 9, and 10 (April 2018) and St. 1, 2, 5, 7, 8, and 10 (June 2018) for environmental factor measurements. Sub-sampling was also conducted at St. 1, 5, 7, 8, and 10 (April 2018) and St. 1, 8, and 10 (June 2018) to analyze nucleocytoplasmic large DNA viruses (NCLDVs) and the eukaryotic plankton community (EPC). Floating drift ice was observed in April (**a**), but not in June (**b**) due to the increase in water temperature.

**Figure 2 microorganisms-11-00169-f002:**
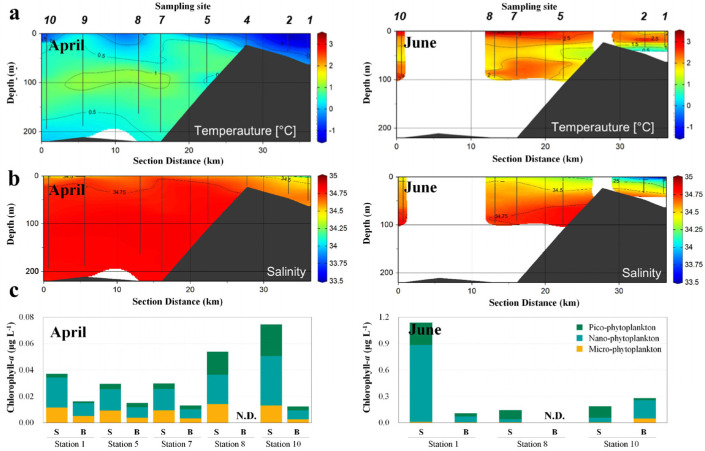
Changes in water temperature (**a**), salinity (**b**), and chlorophyll–*a* concentration (**c**) in the Sub–Arctic Kongsfjorden in April and June 2018. Water temperature and salinity were measured at sites (St.) 1, 2, 4, 5, 7, 8, 9, and 10 in April 2018, and St. 1, 2, 5, 7, 8, and 10 in June 2018. Sub–sampling was also conducted at St. 1, 5, 7, 8, and 10 in April and at St. 1, 8, and 10 in June to measure chlorophyll-*a* concentration. N.D.: No data, S: Surface, B: Bottom.

**Figure 3 microorganisms-11-00169-f003:**
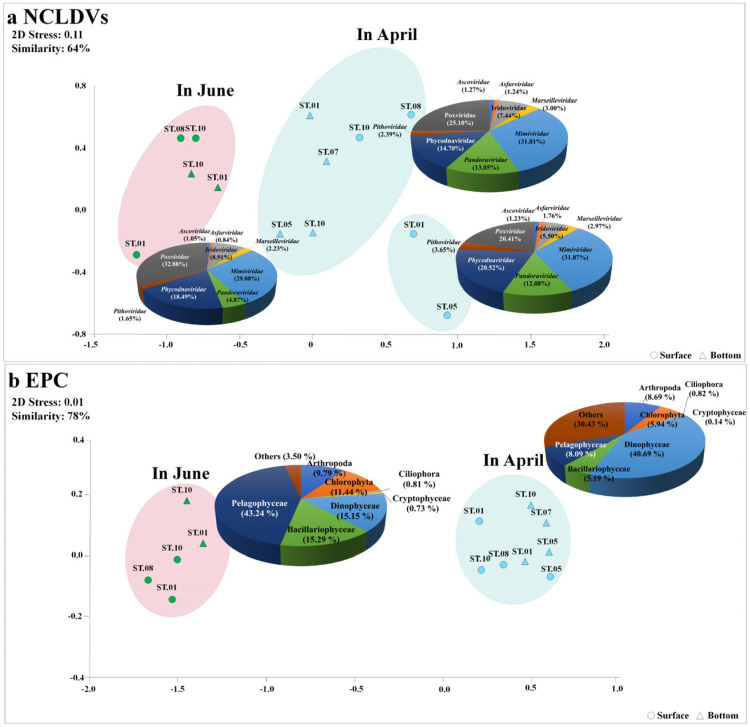
Non–metric multidimensional scaling (NMDS) plot in nucleocytoplasmic large DNA viruses (NCLDVs) (**a**) and the eukaryotic plankton community (EPC) (**b**) in the Sub–Arctic Kongsfjorden in April and June 2018. A total of eight sites were investigated during the cruises. The NMDS plot was analyzed using the Bray–Curtis dissimilarity method. All data were normalized by the fourth roots. The pie charts indicate the high–ranking taxonomic distribution at the family level for NCLDVs and the phylum (or class) level for the EPC.

**Figure 4 microorganisms-11-00169-f004:**
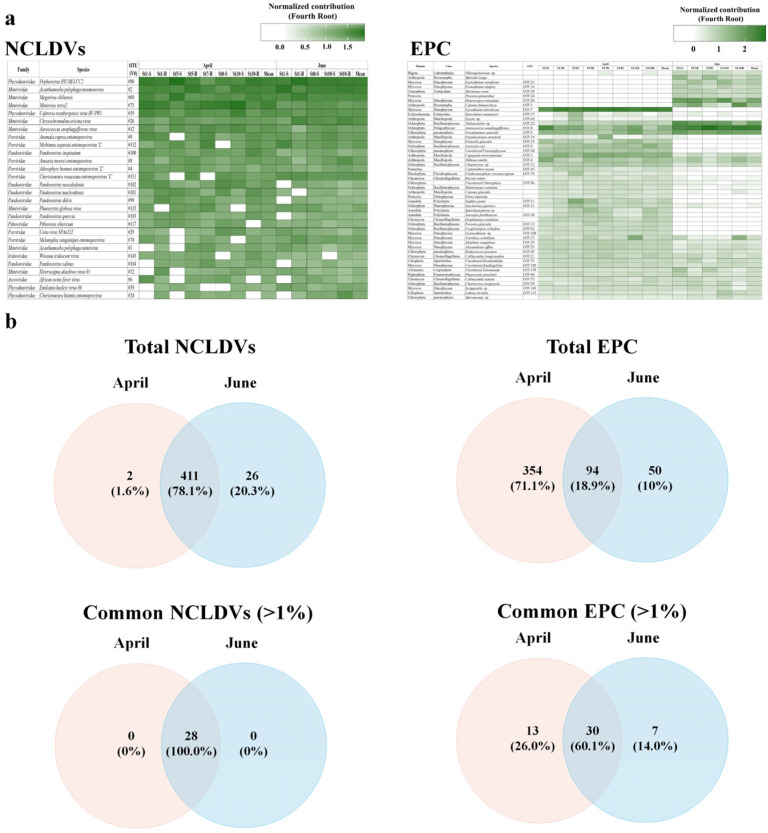
Changes in common nucleocytoplasmic large DNA viruses (NCLDVs; common NCLDV operational taxonomic units (OTUs) at mean relative abundances > 1%), and the eukaryotic plankton community (EPC; common plankton OTUs at the mean relative abundances > 1%) in the Sub–Arctic Kongsfjorden in April and June 2018. (**a**) The heatmap displays the fourth root normalized data, ranging from 0–3. The common OTUs and sampling sites indicate hierarchical agglomerative clustering using the group average with the Bray–Curtis dissimilarity method. (**b**) Venn diagram showing the shared and unique common NCLDVs and EPC OTUs among groups based on NMDS results (see Figure 3). Detailed information on species names of OTU numbers is described in Appendix A.

**Figure 5 microorganisms-11-00169-f005:**
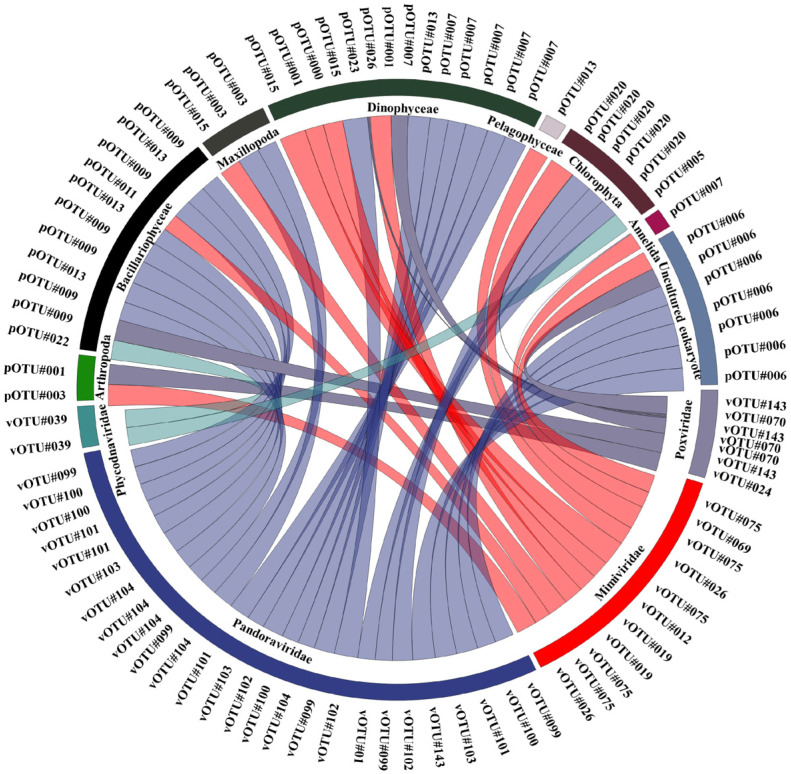
Associations between nucleocytoplasmic large DNA viruses (NCLDVs) and the eukaryotic plankton community (EPC) in the Sub–Arctic Kongsfjorden. Significant pairwise comparisons of the Pearson correlation coefficients between NCLDV and EPC lineages. Detailed information (species names of operational taxonomic unit [OTU] numbers and correlation coefficients) is described in Appendix A.

**Figure 6 microorganisms-11-00169-f006:**
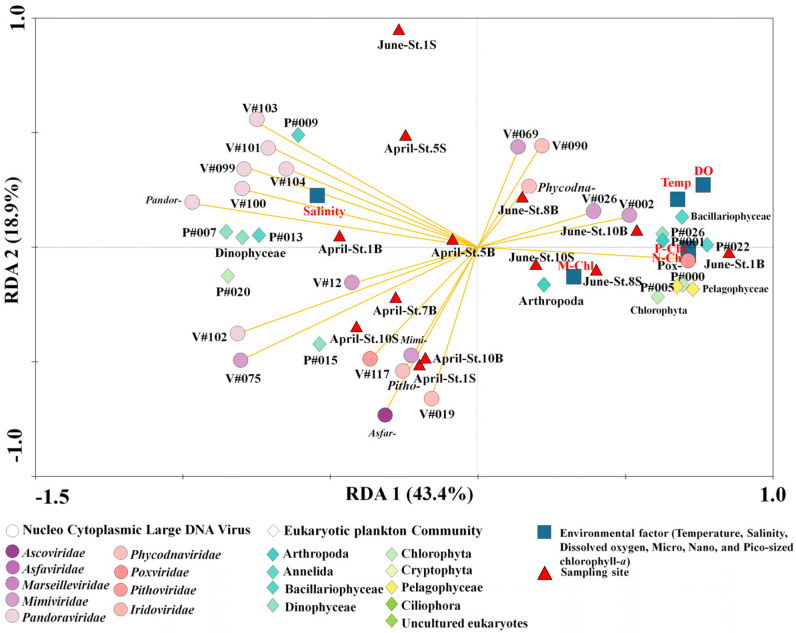
Redundancy analysis (RDA) ordinations of 14 common nucleocytoplasmic large DNA viruses (NCLDVs; common NCLDV operational taxonomic units (vOTUs) and six NCLDVs family selected the mean relative abundances of 1%) analysed using ten common eukaryotic plankton communities (EPCs; common plankton OTUs selected the mean relative abundances of 1%), two phyla, three classe.s, and six environmental factors from 13 sampling sites in the Sub–Arctic Kongsfjorden in April and June 2018. Correlations among the environmental factors (red color), pOTUs of EPC (green color), and NCLDVs vOTUs (orange arrow lines) are shown. The first two RDA axes are represented by the length and angle of the arrows, respectively. Detailed information on species names for each OTU number is described in Appendix A.

**Figure 7 microorganisms-11-00169-f007:**
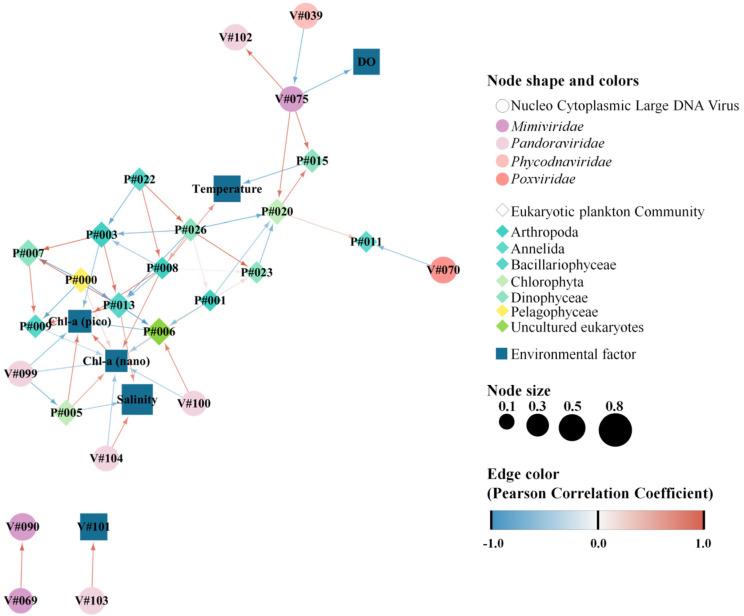
Network analysis showing the co–occurrence between common nucleocytoplasmic large DNA viruses (NCLDVs) and the eukaryotic plankton community (EPC), represented as blue and beige nodes, respectively. Lines between nodes indicate positive (blue) and negative (red) PCC correlation > |0.5| (two–sided pseudo *p*–value < 0.05) between the abundances of linked taxa. Detailed information (species names of operational taxonomic unit [OTU] numbers and correlation coefficients) is described in Appendix A.

## Data Availability

The datasets presented in this study can be found in online repositories. The names of the repository/repositories and accession numbers can be found below: https://www.ncbi.nlm.nih.gov/genbank/, PRJNA848283 accessed on 12 January 2022.

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
