# Peer review of "Covariance of Marine Nucleocytoplasmic Large DNA Viruses with Eukaryotic Plankton Communities in the Sub-Arctic Kongsfjorden Ecosystem: A Metagenomic Analysis of Marine Microbial Ecosystems"

_microorganisms, 2023, doi:10.3390/microorganisms11010169_

Round 1

Reviewer 1 Report

No suggestions.

I found the paper quite interesting and well written

Author Response

Thank you

Reviewer 2 Report

This study investigated the distributions of the nucleocytoplasmic large DNA viruses and the eukaryotic plankton community, and their interplay in the Krossfjorden ecosystem of Svalbard (Norway) using metagenomics and metabarcoding. I find this study is well-written and organized. The methods are described in details. The authors also did lots of statistic analyses. I think the results of this study are valuable for the understanding of the Sub-Arctic Kongsfjorden ecosystem.

Figure 4 is missing.

Author Response

We have described the point by point response as an attached file

Reviewer 3 Report

In this paper, the authors aim to characterize the interplay between NCLDVs and the eukaryotic plankton community (EPC) in the sub-Arctic area in the Krossfjorden ecosystem of Svalbard (Norway) by using metagenomics and metabarcoding in April and 26 June, 2018. They found that NCLDVs and EPC populations were similar between surface and bottom layers, however, differed between April and June sampling points. Unfortunately, however, there are serious problems regarding to the methods, results and conclusions of their study.

1. “2,138 contigs affiliated with NCLDVs were detected via metavirome analysis”. However, more than 50% of contigs are less than 1 kb, and only 14 contigs are longer than 10 kb (table S2). Given the high frequency of horizontal gene transfer among large/giant viruses and their hosts as well as the very large genomic size of NCLDVs, it is almost impossible to accuralty classify NCLDVs only based on sequences < 1 kb. Their results clearly told the truth as well. For example, “Poxviridae” accounted for an essential proportion of the NCLDVs diversity (25% and 33%). Currrently, hosts of poxviruses are vertebrates and invertibrates (mainly insects). Do they infect protists? How to understand the relationship, if there was, between poxviruses and EPC?

2. Lack of data verifying the coverage and specificity of the barcoded primers for 18S rRNA gene.

Author Response

(The authors gave the same response as above.)

Round 2

Reviewer 3 Report

I appreciate the authors for their efforts on revision, while unfortunately, I have not been convinced by their responses. 

Author Response

Point-by-Point Response to Reviewers

Microorganisms

Revised manuscript title: Covariance of Marine Nucleocytoplasmic Large DNA Viruses with Eukaryotic Plankton Communities in the Sub-Arctic Kongsfjorden Ecosystem: A Metagenomic Analysis of Marine Microbial Ecosystems

Manuscript ID: microorganisms-2093632

Prof. Dr. Martin Von Bergen, Editor-in-Chief, Microorganisms

We appreciate the editorial comments and those from the reviewers concerning our manuscript, which were highly helpful and constructive for revising it. We have carefully considered all the comments and made changes accordingly. We have highlighted the revised sentences (or words) in the manuscript in yellow.

Comments of reviewer #3 (RC3)

RC 3-1: I appreciate the authors for their efforts on revision, while unfortunately, I have not been convinced by their responses.

Answer: We apologize for being unable to convince Reviewer 3 with our previous response. I think there are two main reasons for this. The first is the contig size. I agree that a contig size of 500 bp is small in practice. Generally, my research team selects contigs with a size of at least 1 kb or more and uses them as data. However, in this study, we were unable to secure many contigs larger than 1 kb. I analysed two assemblers to figure out what the problem of assembling was, but similar results were obtained when analysing using “MetaSPAdes” and “Megahit” programmes. Therefore, we plan to ask other researchers at our institute to obtain samples and analyse them again to identify the problem. The second is the host ranges of Poxviridae viruses. I addressed this in the last reviewer’s comment. Members of the family Poxviridae can infect a diverse range of vertebrates and invertebrates. In this study, we have focused on the ecological interaction of EPCs and NCLVDs in the sub-Arctic Ocean. Therefore, we do not know exactly which virus infects which host. We are looking at which virus is correlated with multiple appearances of a certain host. We believe that our results in this study (e.g., correlation between Calanus finamchicus and members of Poxviridae) will serve as a basis for future infection experiments.
